# Exceptional long-term sperm storage by a female vertebrate

**Brenna A. Levine[1,2], Gordon W. Schuett[2,3], Warren Booth** [1,2]*

**1** Department of Biological Science, The University of Tulsa, Tulsa, Oklahoma, United States of America,
**2** Chiricahua Desert Museum, Rodeo, New Mexico, United States of America, **3** Department of Biology and
Neuroscience Institute, Georgia State University, Atlanta, Georgia, United States of America

* warren-booth@utulsa.edu

## Abstract

Females of many vertebrate species have the capacity to store sperm within their reproductive tracts for prolonged periods of time. Termed long-term sperm storage, this phenomenon has many important physiological, ecological, and evolutionary implications, particularly to the study of mating systems, including male reproductive success and post-copulatory sexual selection. Reptiles appear particularly predisposed to long-term sperm storage, with records in most major lineages, with a strong emphasis on turtles and squamates (lizards, snakes, but not the amphisbaenians). Because facultative parthenogenesis is a competing hypothesis to explain the production of offspring after prolonged separation from males, the identification of paternal alleles through genetic analysis is essential. However, few studies in snakes have undertaken this. Here, we report on a wild-collected female Western Diamond-backed Rattlesnake, *Crotalus atrox*, maintained in isolation from the time of capture in September 1999, that produced two healthy litters approximately one and six years post capture. Genetic analysis of the 2005 litter, identified paternal contribution in all offspring, thus rejecting facultative parthenogenesis. We conclude that the duration of long-term sperm storage was approximately 6 years (71 months), making this the longest period over which a female vertebrate has been shown to store sperm that resulted in the production of healthy offspring.

## 1. Introduction

The ability for females to store viable spermatozoa in their reproductive tracts, capable of retaining fertilization capacity for months or even years post insemination, has been reported across a variety of vertebrate species [1–7]; a reproductive phenomenon termed long-term sperm storage (LTSS) [1, 8]. The duration of LTSS prior to fertilization varies greatly among taxa [4]. In mammals, sperm typically survives in the uterus for only a few hours to several days, though in bats it may persist for many months [2, 9]. In ectothermic vertebrates, particularly the non-avian reptiles (i.e., chelonians, crocodilians, Rhynchocephalia, and squamates), LTSS is remarkably prolonged, with duration reports of months to several years (reviewed by [1, 2, 4, 8, 10–15]. In snakes, this may be assisted by the presence of anatomical structures,

**Data Availability Statement:** All files, including the Structure and COLONY input files and the custom Python3 script within a Jupyter Notebook, are publicly available on GitHub (https://github.com/brenna-levine/atrox_LTSS).

**Funding:** Funding was provided by a Research Incentive Award, and a Research Creative Activities Award (Arizona State University West) to GWS, and Faculty Startup Funds from The University of Tulsa to WB. Zoo Atlanta and Chiricahua Desert Museum provided additional financial assistance. The funders had no role in study design, data collection and analysis, decision to publish, or preparation of the manuscript.

**Competing interests:** The authors have declared that no competing interests exist.

such as sperm storage tubules in the posterior infundibulum, where spermatozoa migrate post mating [8]. Despite these reports, few studies have applied molecular genetic testing to conclusively confirm the presence of paternal alleles in the resulting offspring. Given the widespread prevalence of facultative parthenogenesis (FP) across the phylogenies of both lizards and snakes [16–19], such testing should be considered essential [15]. In reptiles, LTSS has been genetically confirmed, for example, in turtles [3, 20, 21], lizards [22, 23], and several species of snakes [15, 24]. With an estimated storage duration of at least 67 months, the eastern diamond-backed rattlesnake (*Crotalus adamanteus*) holds the record for the longest genetically confirmed case of LTSS of any vertebrate species [15].

The capacity for, and duration of, LTSS has important implications for mating systems and their analysis [1, 2, 25–27]. This is particularly important when considering male reproductive success and the opportunity for sexual selection [25, 28–30]. Accordingly, polygyny-polyandry and multiple paternity have the potential to be strongly influenced by LTSS [22, 24, 31–33], and its impact on post-copulatory sexual selection, including sperm competition [23, 34, 35] and cryptic female choice [36, 37], should be considered when interpreting results.

Given the prevalence of LTSS across vertebrate species, understanding the duration over which viable spermatozoa can be stored has important implications for the maintenance of genetic diversity, conservation, and management. Here, we report on an exceptional case of LTSS in a female pitviper from the New World, the Western Diamond-backed Rattlesnake, *Crotalus atrox* (Fig 1). Through several lines of evidence, including robust genetic screening using double-digest restriction-site associated DNA (ddRAD) sequencing, we conclusively reject the competing hypothesis of FP [15], an alternative reproductive mode in squamate reptiles [38, 39], and conclude that LTTS persisted for at least 71 months. This finding represents the longest duration of genetically confirmed sperm storage for any vertebrate species which resulted in the production of viable offspring.

## 2. Materials and methods

### 2.1. Subject

The female subject (ID: Ca-149) is a Western Diamond-backed Rattlesnake (*Crotalus atrox*), field collected as an adult from central Arizona on September 17 1999, and maintained in the laboratory (Arizona State University) under strict isolation from the moment of capture (see electronic supplementary material for husbandry conditions). At the time of collection, Ca-149 was deemed to be in excellent health (740 mm snout-vent length, 50 mm tail length, and 435.0 g). On August 22 2000, 340 days after her collection from the wild, Ca-149 gave birth to seven healthy offspring (four males, three females). No infertile ova were present. On August 18 2005, approximately five years (1,822 days) following the first litter, and nearly 71 months (2,162 days) after being collected, Ca-149 produced a second litter of nine offspring (five males, four females). Again, all neonates were healthy in appearance and no infertile ova were present.

### 2.2. Captive care

Ca-149 was maintained in a glass enclosure (91 cm L x 30 cm W x 25 cm H) with a screen cover, supplied with newsprint as a floor covering. Heat tape (8 cm wide), was situated beneath and across the front end of the cage and maintained at 35°C. From the time of collection and during her time in the laboratory Ca-149 was maintained in strictly isolation. Artificial lighting (eight 40 W fluorescent tubes positioned 3 m above the cage) was electronic timer-controlled to a simulate natural photoperiod year round. Pre-killed, thawed laboratory rodents (hamsters and rats) were offered as food every 10 days until 15 November. Water was available in a glass

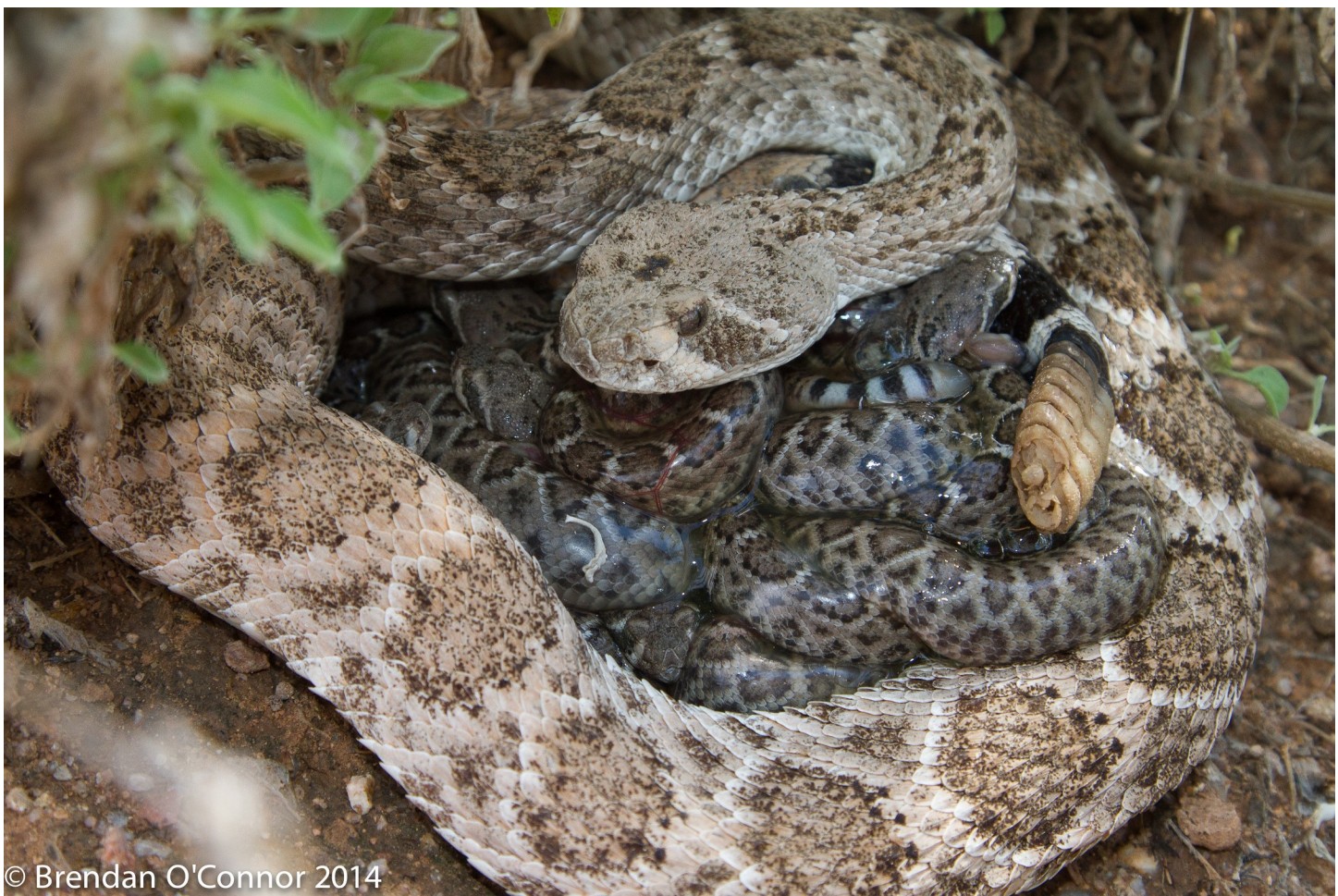

**Fig 1. Female western diamond-backed rattlesnake, *Crotalus atrox*, with newborn offspring.** Photograph courtesy of Brendan O'Connor.

bowl *ad libitum* year-round. From 15 November to 1 March, Ca-149 was maintained under dark and cool conditions to simulate hibernation. Approximately 0.1 ml of blood was collected from 8 of the 9 offspring and female Ca-149 through caudal vein venipuncture, using sterile disposable Tuberculin syringes fitted with 25-G 5/8" needles. Details of collection permits and Institutional Animal Care and Use Committee approvals can be found in the Ethics Statement.

## 2.3. DNA library preparation

Whole genomic DNA was extracted from blood collected from offspring (n = 8 of 9) and female Ca-149 using the Qiagen DNeasy Blood and Tissue Kit (Qiagen), and the concentrations of DNA extracts were quantified with a Qubit 4 Fluorometer (Invitrogen[TM]). The presence of whole genomic DNA was confirmed by separating 5 μL of each DNA sample on a 1.5% agarose gel for 60 minutes at 100 mV and visualizing with SYBR[TM] Safe DNA gel stain (Invitrogen[TM]). DNA was then prepared for high throughput, parallel sequencing using a double-digest restriction-site associated DNA (ddRAD) library preparation protocol [40], slightly modified from that described for snakes by Levine et al. [41]. Approximately 1000 ng DNA of each sample was digested in a thermocycler for 16 hours at 37°C with 1 μL each of FastDigest[TM] restriction enzymes *PstI* and *MspI* (Thermo Scientific[TM]), 5 μL FastDigest 10× buffer

(Thermo Scientific™) and 3 μL molecular grade water. It was confirmed that DNA was digested by separating a 5 μL aliquot of each digest on a 1.5% agarose gel and visualizing with SYBR™ Safe gel stain (Invitrogen™). Digests were cleaned using 1.5× Serapure solution (as prepared from Sera-Mag SpeedBeads™ [GE Healthcare] following the protocol of Rohland and Reich [42]).

A unique 5-base pair (bp) barcode was ligated to the *PstI* overhangs for each sample using a reaction mix of 2 μL barcoded P1 adapter, 2 μL P2 adapter, 3 μL 10× T4 ligase buffer (New England BioLabs® Inc.), and 1 μL T4 ligase (New England BioLabs® Inc.). The reaction mix was incubated in a thermocycler with the following thermal profile: 22˚C for 60 minutes, 65˚C for 10 minutes with a -1˚C/minute ramp rate, and a 20˚C hold. Barcoded samples were then pooled and an additional clean-up with 1.5× Serapure solution was performed. The pooled library was size-selected using a Blue Pippin (Sage Science), retaining only those fragments 300 bp ± 50 bp in length. This size-selection range was chosen by performing an *in silico* digestion of the published prairie rattlesnake, *C. viridis viridis*, genome [43], with the cut-site sequences of *PstI* (CTGCA^G) and *MspI* (C^CGG) using the program *FRAGMATIC* [44].

Finally, four Phusion™ PCRs were performed to reduce the introduction of PCR error into the library. Each reaction mix contained 5 μL size-selected library, 5.8 μL molecular grade water, 0.5 μL MgCl$_2$ (50 mM), 0.5 μL dNTPs (10 mM), 2 μL PCR 1 Primer (2 μM), 2 μL indexed PCR 2 primer (2 μM), 4 μL 5× Phusion™ HF Buffer (Thermo Scientific), and 0.2 μL Phusion™ DNA polymerase (Thermo Scientific). Reaction mixes were incubated in a thermocycler with the following temperature profile: 1 cycle of 98˚C for 1 minute; 10 cycles of 98˚C for 15 seconds, 62˚C for 30 seconds, and 72˚C for 30 seconds; 1 cycle of 72˚C for 7 minutes; 20˚C hold. The four PCR products were subsequently pooled and cleaned with 1.5× Serapure solution prior to Illumina sequencing. Single-end 100-bp sequencing was performed on an Illumina HiSeq4000 at the University of Oregon Genomics and Cell Characterization Core Facility (GC3F).

## 2.4. Bioinformatics

Post-sequencing, raw fastq files were inspected for quality using FastQC [45]. The process_-radtags module of program Stacks v. 2.41 [46, 47] was then used to clean and demultiplex the raw sequencing reads. Raw reads were clustered into loci via both reference-based and *de novo* analysis pipelines in Stacks v. 2.41 [48] to confirm that the pipeline didn't impact the clustering of reads into loci. Prior to executing the reference-aligned Stacks pipeline, the cleaned sequencing reads were aligned to the reference genome sequence for the *C. v. viridis* [43] using the BWA-MEM algorithm implemented in BWA [49], with default settings. The *gstacks* module of Stacks v. 2.41 was then executed to identify and genotype single nucleotide polymorphisms (SNP) at each locus, with the alpha thresholds for discovering SNPs *(—var-alpha)* and calling genotypes *(—gt-alpha)* set more stringent than the default values (= 0.001).

For *de novo* clustering and genotyping of SNPs, combinations of Stacks core parameters (*m*, *M*, and *n*) were compared to identify the parameter combination at which the number of polymorphic loci shared by at least 80% of samples stabilized [48]. Values of *M* = *n* from 1–9 were tested, while holding *m* at 3. After identifying the parameters at which the number of polymorphic loci shared across samples stabilized (*M* = *n* = 3), the *denovo_map.pl* wrapper script was executed to identify and genotype variable loci in all individuals. As for the reference-aligned analysis, the alpha threshold for discovering SNPs and calling genotypes was set to 0.001.

For both reference-based and de novo analyses, the populations module was used to identity variable loci present in all individuals (r = 1.0) and to produce output formatted as a Structure file, while retaining only the first SNP at each locus with the—write_single_snp option.

## 2.5. Scan for paternal alleles and sibship reconstruction

To determine whether male offspring resulted from LTSS or FP, a custom Python3 script was written to scan male genotypes in each Structure file for putative paternal alleles. Prior to running the script, both the reference-based and *de novo* Structure files were modified to discard genotypes for female offspring while retaining those for Ca-149 and all male offspring. The script then identified loci at which the mother was homozygous and cross-referenced male alleles against those of the homozygous female to identify males with alleles that differed from the maternal alleles at each locus. Since multiple sources of error can result in heterozygotes called as homozygotes and vice versa [50], the number of loci at which all males (n = 5) displayed alleles that differed from those of the mother was summed to yield a conservative estimate of the frequency of paternal alleles in the male offspring.

To supplement the above analysis, and to determine the number of sires that may have contributed to the litter, relationships among the offspring were also inferred via sibship reconstruction with COLONY v. 2.0.6.6 [51]. A subset of the *de novo* loci, as filtered via program PLINK v. 1.9 beta [52] to retain only those with a minor allele frequency (MAF) of 0.5, was analyzed. Five iterations of medium length runs were conducted. Parameters for the COLONY runs included: male and female polygamy, no inbreeding, no sibship size prior, no known or candidate parents, and a genotyping error rate of 0.05 per SNP.

## 3. Results

The reference-aligned Stacks analysis identified 3,493 SNPs present in all individuals at a mean effective per-sample coverage of 29.0×, whereas the de novo analysis pipeline identified 4,228 SNPs present in all individuals at a mean effective per-sample coverage of 41.1×. With respect to the reference-aligned data, of 1,197 loci at which the mother was inferred to be homozygous, putative paternal alleles were found in all male offspring at 61 loci (5.1%). With respect to the de novo clustered loci, all males displayed paternal alleles at 67 of the 1,445 loci for which the mother was homozygous (4.6%). All loci for which the mother was homozygous yet males displayed putative paternal alleles are found within Python3 Jupyter notebook (https://github.com/brenna-levine/atrox_LTSS). Supporting these results demonstrating the presence of paternal alleles in male offspring, COLONY assigned all offspring to one full-sibship when a subset of high MAF loci were analyzed (n = 417).

## 4. Discussion

Combining captive history and genomic screening, we unequivocally identify LTSS as the reproductive mechanism underlying the production of offspring from female Ca-149, following a period of approximately 71 months of isolation post capture. With the identification of paternal alleles in all offspring, we also conclusively reject FP as an alternative developmental mechanism for any of the male offspring. All offspring analyzed were sired by a single male. While offspring from the first birth were unavailable for genomic screening, the captive history of this female shows that not only can female *C. atrox* store sperm over prolonged periods of time, they can also utilize this sperm over multiple distinct reproductive events.

Reviewed in Booth & Schuett [15, 17], caenophidian snakes (advanced snakes including the pitvipers) exhibit common characteristics indicative of FP that differ markedly from LTSS. Specifically, a high number of infertile ova, male-only offspring due to the possession of ZZ/ZW sex chromosomes (males are ZZ and WW females are not viable), and offspring exhibiting developmental abnormalities, are commonly observed in instances of FP. In contrast, litters genetically confirmed as LTSS in pitvipers, have been indistinguishable from those produced without sperm storage (results presented here, [15]). Given these distinct differences in litter

characteristics, the longest reported case that is attributed to LTSS (84 months in the Javan File Snake, *Acrochordus javanicus*, [53]), cannot be conclusively confirmed as LTSS, as the litter characteristics are also strongly indicative of FP [15, 17]. Furthermore, it was believed that the embryo died during development, shortly prior to the death of the mother. Thus, this frequently cited case failed to produce a viable offspring. Therefore, to our knowledge, the duration of successful LTSS reported here, which can be conclusively disentangled from FP, represents the longest confirmed duration over which any vertebrate has been found to stored sperm that was later used in the production of viable offspring.

Sperm storage by females has the potential to profoundly affect mating systems, particularly male reproductive success and the strength of post-copulatory sexual selection [25, 29–33, 54]. In *C. atrox*, while the species exhibits characteristics that could promote male-biased sexual selection (e.g., male-male combat for access to females, male-biased sex ratio, and male-biased sexual size dimorphism), Levine et al. [30] found no such pattern. In the population studied by Levine et al. [30], high instances of multiple paternity were previously reported [33]. Furthermore, of 12 females for which multiple paternity litters were identified, 10 were radio-tracked over the mating season with both courtship and mating behaviors reported. Of these, seven were not observed to mate with multiple males [33]. Thus, while mating events with additional males may have been missed, LTSS from a previous season cannot be excluded.

In snakes, while there is limited evidence that sperm stored over multiple reproductive seasons can compete with spermatozoa from recent inseminations [1, 12, 24, 32, 55], general models show that overlapping ejaculates resulting in multiple paternity can erode the strength of post-copulatory sexual selection in males [1, 25, 54]. In studies investigating post-mating sexual selection, a primary aim is to measure the opportunity for sexual selection [25, 54], or selection intensity acting directly on mate numbers using Bateman gradients or traits such as body size [28–30, 33, 54, 56, 57]. These types of evolutionary analyses would clearly profit from a broader understanding of LTSS in populations given the eroding effect it may have on sexual selection [24, 30].

While the evolutionary significance of LTSS in regards to its impact on mating systems and its potential to retain genetic diversity previously considered lost from a population can be evaluated theoretically, the physiological mechanism(s) permitting LTSS are at present still unclear. However, anatomical studies have shown that seminal receptacles [8, 58–61], termed "sperm storage tubules", located in the posterior infundibulum of the oviduct, are present in multiple species of crotaline snakes, including *C. atrox*. Post mating, spermatozoa ascend the reproductive tract until reaching these, and sperm is believed to be stored until ovulation. Similarly, uterine muscular twisting has been reported in viperids and also could play a role [60–65]. After copulation, uterine contractions may result in a twisting of the oviducts, with untwisting and therefore sperm migration occurring only after ovulation [63]. In the Neotropical rattlesnake, *C. durissus*, uterine muscular twisting was observed to occur from the formation of coils by the inner layers of the oviducts at the utero-vaginal junction, not actual rotation of the oviducts around their axis [64]. Within these coils, furrows formed through the process of contraction may then retain sperm within this region [63]. Interestingly, while uterine muscular twisting has been observed in both non-vitellogenic and vitellogenic females, as might be expected if it serves a sperm storage function, it has also been observed in gravid females [61, 65], questioning the anatomical function of this structure in LTSS. However, the finding presented here of multiple litters resulting from LTSS suggests that this mechanism is potentially functional in LTSS and retained until all spermatozoa are depleted; a process which may span multiple seasons. Further research into the function of uterine-muscular twisting in LTSS is clearly warranted.

Although the present example of LTTS appears to be exceptional, more frequent testing in the future will likely determine that similar cases are not uncommon in certain taxa of snakes,

and LTSS is potentially more widespread among squamate reptiles than currently known. Such findings would have significant implications for the understanding of mating systems in natural populations, and the evolution of sexual selection. Furthermore, based on these results, rattlesnakes may represent an ideal model system for understanding the mechanisms by which viable sperm can be stored under non-cryogenic conditions [66], the application of which would have significance to assisted reproduction in humans, livestock, and species of conservation concern.

## Acknowledgments

We thank Ryan Sawby, Dale DeNardo, Roger Repp, and Emily Taylor for assistance in the field. We thank the University of Oklahoma Supercomputing Center for Education and Research (OSCER) for access to computational resources.

## Author Contributions

**Conceptualization:** Gordon W. Schuett, Warren Booth.

**Data curation:** Brenna A. Levine, Gordon W. Schuett.

**Formal analysis:** Brenna A. Levine.

**Funding acquisition:** Gordon W. Schuett, Warren Booth.

**Investigation:** Brenna A. Levine, Gordon W. Schuett, Warren Booth.

**Methodology:** Brenna A. Levine, Gordon W. Schuett, Warren Booth.

**Project administration:** Warren Booth.

**Resources:** Warren Booth.

**Writing – original draft:** Brenna A. Levine, Gordon W. Schuett, Warren Booth.

**Writing – review & editing:** Brenna A. Levine, Gordon W. Schuett, Warren Booth.

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
