## [Decision Letter · Decision Letter 0]

26 Apr 2021

PONE-D-21-10790

Exceptional long-term sperm storage by a female vertebrate

PLOS ONE

Dear Dr. Booth,

Thank you for submitting your manuscript to PLOS ONE. After careful consideration, we feel that it has merit but does not fully meet PLOS ONE’s publication criteria as it currently stands. Therefore, we invite you to submit a revised version of the manuscript that addresses the points raised during the review process.

This is an interesting observation on an exceptionally long storage period of sperm in a snake. I agree with the two reviewers that the data are interesting and that the manuscript is well written. I do support their recommendations and comments and request minor revision of the paper.

We look forward to receiving your revised manuscript.

Kind regards,

Stefan Schlatt

Academic Editor

PLOS ONE

Journal Requirements:

2.  We note that Figure 1 in your submission contain copyrighted images. All PLOS content is published under the Creative Commons Attribution License (CC BY 4.0), which means that the manuscript, images, and Supporting Information files will be freely available online, and any third party is permitted to access, download, copy, distribute, and use these materials in any way, even commercially, with proper attribution. For more information, see our copyright guidelines: http://journals.plos.org/plosone/s/licenses-and-copyright.

2.1.    You may seek permission from the original copyright holder of Figure 1 to publish the content specifically under the CC BY 4.0 license.

2.2.    If you are unable to obtain permission from the original copyright holder to publish these figures under the CC BY 4.0 license or if the copyright holder’s requirements are incompatible with the CC BY 4.0 license, please either i) remove the figure or ii) supply a replacement figure that complies with the CC BY 4.0 license. Please check copyright information on all replacement figures and update the figure caption with source information. If applicable, please specify in the figure caption text when a figure is similar but not identical to the original image and is therefore for illustrative purposes only.

Reviewers' comments:

Reviewer's Responses to Questions

**Comments to the Author**

1. Is the manuscript technically sound, and do the data support the conclusions?

Reviewer #1: Yes

Reviewer #2: Yes

2. Has the statistical analysis been performed appropriately and rigorously? 

Reviewer #1: Yes

Reviewer #2: N/A

3. Have the authors made all data underlying the findings in their manuscript fully available?

Reviewer #1: Yes

Reviewer #2: Yes

4. Is the manuscript presented in an intelligible fashion and written in standard English?

Reviewer #1: Yes

Reviewer #2: Yes

5. Review Comments to the Author

Reviewer #1: The authors report on a wild-collected female Crotalus atrox, maintained in isolation from 1999, that produced two healthy litters approximately one and six years post capture. Genetic analysis of the second litter identified paternal contribution in all offspring, thus rejecting facultative parthenogenesis. They suggest the long-term sperm storage to last approximately 71 months, making it the longest period over which a female vertebrate has been shown to store sperm that resulted in the production of healthy offspring from LTSS.

This is an observational study reproting an interesting finding with implications for sperm competition, mating and reproductive strategies in a snake. The methods applied and the data obtained appear valid. I have only a very few minor points that have tob e clarified during a revision oft he paper.

Material and methods:

I am missing a note on the approval for the collection of the animal from the wild as well as a licence for keeping it in captivity.

Whole genomic DNA was obtained from offspring: Please explain how this was done: From which tissues? Were the animals killed? By which means? And, again, under which licence?

Discussion:

Appears rather long. Please shorten if possible.

Reviewer #2: This is manuscript is well written and provides unprecedented data on exceptional long-term sperm storage for 71 months in a female Western Diamond-backed Rattlesnake (Crotalus atrox). The methods are soundly carried out. Descriptions are detailed and well documented. The combination of data obtained in captivity, genetic screening and the observation of paternal alleles in all offspring showed a very accurate result, confirming the differentiation between long-term sperm storage (LTSS) and facultative parthenogenesis (FP). Both males and females were born, all in good condition and no anomalies present, reinforcing the results presented in the paper. Some inttriguing questions about long-term sperm storage remain uncertain: we still need to understand how spermatozoa are able to survive during this period and how females may produce multiple clutches using these stored sperm. The authors of this study are experts on this subject and this manuscript is a huge contribution to the knowledge on the reproductive biology of vertebrates. This is a good paper and I look forward to see this paper published. Minor suggestions are included below:

INTRODUCTION

Line 27-29: The authors mention in these lines that ectothermic vertebrates (non-avian reptiles) may show sperm storage for several years, however they do not mention where sperm is stored. A brief anatomical description of sperm storage sites in the oviducts of female reptiles, mainly for snakes is important. Later, the authors mention in the discussion section, sperm storage in UMT (uterine muscular twisting) in the útero-vaginal junction and in the infundibular portion. The inclusion of this information in the introduction will contribute to a general view on sperm storage in the oviduct, contributing to a better understanding of this reproductive tactic by the readers.

MATERIAL AND METHODS

The dataset and analysis presented by the authors strongly support the goals of the study, reflecting huge experience of the authors within these techniques. The methodology is well detailed and the replication of this study is possible. On line supporting information contains the protocols and algorithms used in this study, allowing their application in future research.

Line 121: Please, include the meaning of SNPs in the first time that this abbreviation appears in the text.

DISCUSSION

This section is short, however interesting and well written. However, I suggest the inclusion of some recent publications on uterine muscular twisting (UMT) and infundibular sperm storage in Viperidae snakes to improve this section.

Line 217-219: This paragraph may be improved with the inclusion of these recent publications (Silva et al 2019; Silva et al., 2020, Barros et al., 2020; Souza and Almeida-Santos, 2020) on sperm storage in Bothrops and Lachesis (closely related taxa). Recent data show that the UMT does not prevent the passage of spermatozoa to the infundibulum after mating as it previously hypothesized for some Viperidae snakes. The UMT has been described in vitellogenic, non-vitellogenic and pregnant females, with and without the presence of sperm, raising many doubts about the function of this structure. Recent findings showed that sperm storage in Bothrops and Lachesis occur in the furrows of the útero-vaginal junction or in the UMT, and in infundibular sperm receptacles. It suggests that Crotalus may also have infundibular sperm receptacles. Please, find the references for the publications below:

SILVA, KARINA M. P.; BARROS, VERÔNICA A.; ROJAS, CLAUDIO A.; ALMEIDA'SANTOS, SELMA M. .Infundibular sperm storage and uterine muscular twisting in the Amazonian lancehead,. Anatomical Record-Advances in Integrative Anatomy and Evolutionary Biology, v. 2019, p. ar.24309-1-10, 2019

BARROS, VERÔNICA A.; ROJAS, CLAUDIO A; ALMEIDA'SANTOS, SELMA M Biologia Reprodutiva das Serpentes Jararacas: Ciclos e Comportamentos, Dimorfismo e Maturidade sexual. 1. ed. Ilhas Mauricio: Novas Edições Acadêmicas, 2020. v. 1. 173p

SILVA, KARINA M.P.; BRAZ, HENRIQUE B.; KASPEROVICZUS, KARINA N.; MARQUES, OTAVIO A.V. ; ALMEIDA-SANTOS, SELMA M. Reproduction in the pitviper Bothrops jararacussu: large females increase their reproductive output while small males increase their potential to mate. Zoology, v. xx, p. 125816, 2020

SOUZA, ELETRA; ALMEIDA'SANTOS, SELMA MARIA. Reproduction in the bushmaster ( Lachesis muta ): Uterine muscular coiling and female sperm storage. Acta Zoologica ONLINE, v. 2020, p. 1-12, 2020

REFERENCES

This reference was not cited in the manusccript

52 -Purcell S, Neale B, Todd-Brown K, Thomas L, Ferreira MAR, Bender D, Maller J, Sklar P, de Bakker PIW, Daly MJ, Sham PC. 2007 PLINK: a toolset for whole-genome association and population-based linkage analysis. A. J. Hum. Genet. 81, 559–575.

6. PLOS authors have the option to publish the peer review history of their article (what does this mean?). If published, this will include your full peer review and any attached files.

Reviewer #1: No

Reviewer #2: No

---

## [Author Response · Author response to Decision Letter 0]

30 Apr 2021

Response to Reviewer comments:

We thank the reviewers for the constructive feedback and have taken these into account in our revision of the manuscript. These revisions, albeit minor, enhance the quality of the manuscript. 

Reviewer #1: The authors report on a wild-collected female Crotalus atrox, maintained in isolation from 1999, that produced two healthy litters approximately one and six years post capture. Genetic analysis of the second litter identified paternal contribution in all offspring, thus rejecting facultative parthenogenesis. They suggest the long-term sperm storage to last approximately 71 months, making it the longest period over which a female vertebrate has been shown to store sperm that resulted in the production of healthy offspring from LTSS.

This is an observational study reporting an interesting finding with implications for sperm competition, mating and reproductive strategies in a snake. The methods applied and the data obtained appear valid. I have only a very few minor points that have to be clarified during a revision of the paper.

Material and methods:

I am missing a note on the approval for the collection of the animal from the wild as well as a licence for keeping it in captivity.

Response – Information regarding the collection permits and IACUC approval are already provided in the Ethics Statement section of the manuscript. 

Whole genomic DNA was obtained from offspring: Please explain how this was done. From which tissues? Were the animals killed? By which means? And, again, under which license? 

Response – Relevant information has been included in the methods section and the lACUC protocol number can be found in the Ethics Statement.

Discussion:

Appears rather long. Please shorten if possible. 

Response – While reviewer 1 suggests shortening the discussion, reviewer 2 does not, and but instead suggests expansion of the section discussing the anatomical mechanisms of sperm storage in snakes. In light of recent studies which report uterine muscular twisting in gravid females (a finding which contradicts the function of this anatomical process in LTSS), our report of sequential LTSS litters despite isolation from males suggest that this mechanism is indeed important in LTSS, and may actually permit the storage of sperm over multiple reproductive events. As such, we feel that this, in concert with the significance of our findings to sexual selection, justifies following the recommendation of reviewer 2.

Reviewer #2: This is manuscript is well written and provides unprecedented data on exceptional long-term sperm storage for 71 months in a female Western Diamond-backed Rattlesnake (Crotalus atrox). The methods are soundly carried out. Descriptions are detailed and well documented. The combination of data obtained in captivity, genetic screening and the observation of paternal alleles in all offspring showed a very accurate result, confirming the differentiation between long-term sperm storage (LTSS) and facultative parthenogenesis (FP). Both males and females were born, all in good condition and no anomalies present, reinforcing the results presented in the paper. Some intriguing questions about long-term sperm storage remain uncertain: we still need to understand how spermatozoa are able to survive during this period and how females may produce multiple clutches using these stored sperm. The authors of this study are experts on this subject and this manuscript is a huge contribution to the knowledge on the reproductive biology of vertebrates. This is a good paper and I look forward to see this paper published. Minor suggestions are included below:

INTRODUCTION

Line 27-29: The authors mention in these lines that ectothermic vertebrates (non-avian reptiles) may show sperm storage for several years, however they do not mention where sperm is stored. A brief anatomical description of sperm storage sites in the oviducts of female reptiles, mainly for snakes is important. Later, the authors mention in the discussion section, sperm storage in UMT (uterine muscular twisting) in the útero-vaginal junction and in the infundibular portion. The inclusion of this information in the introduction will contribute to a general view on sperm storage in the oviduct, contributing to a better understanding of this reproductive tactic by the readers.

Response – a brief statement has been added, which is expanded in the discussion.

MATERIAL AND METHODS

The dataset and analysis presented by the authors strongly support the goals of the study, reflecting huge experience of the authors within these techniques. The methodology is well detailed and the replication of this study is possible. On line supporting information contains the protocols and algorithms used in this study, allowing their application in future research.

Line 121: Please, include the meaning of SNPs in the first time that this abbreviation appears in the text.

Response - done

DISCUSSION

This section is short, however interesting and well written. However, I suggest the inclusion of some recent publications on uterine muscular twisting (UMT) and infundibular sperm storage in Viperidae snakes to improve this section.

Line 217-219: This paragraph may be improved with the inclusion of these recent publications (Silva et al 2019; Silva et al., 2020, Barros et al., 2020; Souza and Almeida-Santos, 2020) on sperm storage in Bothrops and Lachesis (closely related taxa). Recent data show that the UMT does not prevent the passage of spermatozoa to the infundibulum after mating as it previously hypothesized for some Viperidae snakes. The UMT has been described in vitellogenic, non-vitellogenic and pregnant females, with and without the presence of sperm, raising many doubts about the function of this structure. Recent findings showed that sperm storage in Bothrops and Lachesis occur in the furrows of the útero-vaginal junction or in the UMT, and in infundibular sperm receptacles. It suggests that Crotalus may also have infundibular sperm receptacles. 

Response – The section has been expanded and additional citations included.

REFERENCES

This reference was not cited in the manuscript

52 -Purcell S, Neale B, Todd-Brown K, Thomas L, Ferreira MAR, Bender D, Maller J, Sklar P, de Bakker PIW, Daly MJ, Sham PC. 2007 PLINK: a toolset for whole-genome association and population-based linkage analysis. A. J. Hum. Genet. 81, 559–575.

Response – Incorrectly numbers as 47, but now corrected.

---

## [Decision Letter · Decision Letter 1]

10 May 2021

Exceptional long-term sperm storage by a female vertebrate

PONE-D-21-10790R1

Dear Dr. Booth,

We’re pleased to inform you that your manuscript has been judged scientifically suitable for publication and will be formally accepted for publication once it meets all outstanding technical requirements.

Kind regards,

Stefan Schlatt

Academic Editor

PLOS ONE

Additional Editor Comments (optional):

Reviewers' comments:

Reviewer's Responses to Questions

**Comments to the Author**

1. If the authors have adequately addressed your comments raised in a previous round of review and you feel that this manuscript is now acceptable for publication, you may indicate that here to bypass the “Comments to the Author” section, enter your conflict of interest statement in the “Confidential to Editor” section, and submit your "Accept" recommendation.

Reviewer #1: All comments have been addressed

Reviewer #2: All comments have been addressed

2. Is the manuscript technically sound, and do the data support the conclusions?

Reviewer #1: (No Response)

Reviewer #2: Yes

3. Has the statistical analysis been performed appropriately and rigorously? 

Reviewer #1: (No Response)

Reviewer #2: Yes

4. Have the authors made all data underlying the findings in their manuscript fully available?

Reviewer #1: (No Response)

Reviewer #2: Yes

5. Is the manuscript presented in an intelligible fashion and written in standard English?

Reviewer #1: (No Response)

Reviewer #2: Yes

6. Review Comments to the Author

Reviewer #1: (No Response)

Reviewer #2: The authors of the present study “Exceptional long-term sperm storage by a female vertebrate” followed all the sugestions indicated in the review and especially regarding the inclusion of more recent hypotheses about the long-term sperm storage (LTSS) in crotaline snakes. The insertion of this information aimed to improve the quality of the manuscript from the morphophysiological point of view. The study is of high quality and will stimulate new research on the storage of sperm in vertebrates. Minor sugestions are included below:

References

Line 144: change Souz by “Souza” and Almeida Santo by “Almeida-Santos”

7. PLOS authors have the option to publish the peer review history of their article (what does this mean?). If published, this will include your full peer review and any attached files.

Reviewer #1: No

Reviewer #2: No

---

## [Editor Report · Acceptance letter]

26 May 2021

PONE-D-21-10790R1 

Exceptional long-term sperm storage by a female vertebrate 

Dear Dr. Booth:

I'm pleased to inform you that your manuscript has been deemed suitable for publication in PLOS ONE. Congratulations! Your manuscript is now with our production department. 

Kind regards, 

on behalf of

Dr. Stefan Schlatt 

Academic Editor

PLOS ONE